# Blockchain Payment Services in the Hospitality Sector: The Mediating Role of Data Security on Utilisation Efficiency of the Customer

Ankit Dhiraj [1] , Sanjeev Kumar [1], Divya Rani [2] , Simon Grima [3,4,*] and Kiran Sood [5]

1 School of Hotel Management and Tourism, Lovely Professional University, Phagwara 144001, Punjab, India; ankitdhiraj@gmail.com (A.D.); kumarsharma12360@gmail.com (S.K.)
2 Department of Economics, Patliputra University, Patna 800020, Bihar, India; divyaranitargir@gmail.com
3 Faculty of Business, Management and Economics, University of Latvia, LV-1586 Riga, Latvia
4 Department of Insurance, Faculty of Economics Management and Accountancy, University of Malta, MSD 2080 Msida, Malta
5 Chitkara Business School, Chitkara University, Rajpura 140401, Punjab, India; kiran.sood@chitkara.edu.in
* Correspondence: simon.grima@um.edu.mt or simon.grima@lu.lv

**Abstract:** Blockchain technology has the potential to completely transform the hospitality sector by offering a safe, open, and effective method of payment. Increased customer utilisation efficiency may result from this. This study looks into how blockchain payment methods affect hotel customers' intentions to stay loyal by devising four hypotheses. A questionnaire was specifically created and self-administered for this study as a data-gathering tool and distributed to hotel customers. The I.B.M. SPSS and Amos software packages were used to analyse the data of the 301 valid responses. Findings show that hospitality customers may use blockchain payment services if the customer is satisfied with the data security of this payment system. The study also highlighted that customer data security mediated the association between utilisation efficiency and blockchain payment systems. Blockchain payment services can affect visitors' intentions to stay loyal by impacting data security and consumer happiness. Results suggest that blockchain payment systems can be useful for hospitality firms looking to increase client utilisation efficiency. Blockchain can simplify visitor booking and payment processes by providing a safe, open, and effective transacting method. This may result in a satisfying encounter that visitors are more inclined to recall and repeat.

**Keywords:** blockchain; payment services; hotel industry; utilisation efficiency; data; security

## 1. Introduction

As we move through the industry 4.0 and digital transformation age, a significant digital revolution is happening worldwide. As a result, organisations need to evolve to survive. One method to achieve this is utilising cutting-edge technology like IoT, A.I., cloud computing, and blockchain. Blockchain technology has become increasingly important for many nations, entities, and organisations since it offers a novel solution to address the system's inefficiencies. Numerous nations, including the United Arab Emirates [1], the United States [2], Australia [3], Estonia [4], Singapore [5], China [2], Georgia [6], and others, have begun experimenting with or implementing this technology at the production services level, with blockchain underpinning digital currencies. Also, countries like El Salvador have made bitcoin a legal tender [7,8].

Table 1 describes the characteristics of blockchain. The blockchain's immutability results from the fact that new data can be attached but the chain's old data is kept unchanged. Because everyone has access to the same data, blockchain can aid in establishing transparency in the processes. Its fundamental drawback continues to be the lack of flexibility and limited programmability. Decentralisation is one of the fundamental characteristics of the blockchain, which means that data (transactions) or code are kept identical on several

computers, or "nodes", throughout the network. The level of anonymity on a blockchain largely depends on its configuration (public vs. private) and is not a significant concern in many commercial applications with a well-defined group of participants. The process of reaching a consensus over the legitimacy of transactions and determining which entities are permitted to add data is another crucial component of a decentralised network.

**Table 1.** Characteristics of Blockchain.

| Characteristics | Explanation |
| --- | --- |
| Transparency | A limited number of users have access to the data on a blockchain. In particular, they all have the same perspective on facts. |
| Decentralisation | Blockchain technologies are decentralised and do not require a single point of control. Consensus protocols outline how scattered parties can agree on what information should be recorded on a blockchain and the current state of reality. |
| Immutability | Unless a specific portion of the network (for example, the majority of the hashing power in bitcoin) decides to do so, data in a blockchain cannot be changed. It is simple to detect whether data has been altered. |
| Anonymity | In a blockchain, the visibility of identifying information varies from complete anonymity to full identity. |
| Programmability | Blockchains that can be programmed allow for rules (commonly called "smart contracts") that are automatically carried out when certain circumstances are met. |
| Consensus | An agreement component is applied to accomplish settlement on the condition of an organisation, including the legitimacy of exchanges and how choices can be made. |

Source: (adapted from Treiblmaier, 2020) [9].

Building customer-based value propositions has become possible thanks to technological advancement and digitisation in the travel sector. These ideas centre on decentralised autonomous value chains, information transparency, and flexible customisation. Therefore, a paradigm shift from conventional business models to customer-centric ones is required [8]. Preceding coronavirus, travel and the travel industry had formed into perhaps the main monetary area on the planet, supporting more than 320 million positions and giving 10% of the worldwide gross domestic product [6,9,10]. Worldwide the travel industry income is not supposed to arrive at 2019 levels until 2023. In this current year, until April 2023, travellers increased by more than 65%, as per new I.M.F. research on the travel industry in a post-pandemic world.

In contrast, following the financial crisis and the SARS outbreak, the increase was only 8 per cent and 17 per cent, respectively. In the first quarter of 2023, foreign arrivals were already around 80% of the pre-pandemic levels. Over twice as many visitors as in the same period in 2022 travelled abroad in the first three months, according to estimates of 235 million travellers. Nearly 7.8 billion passengers will travel by air by 2036. Like other industries, the hospitality sector had a market worth USD 500 trillion in 2018 and is expected to triple by 2030 [11] (vide Figure 1).

Therefore, it is important to preserve trust between tourists and tourism and hospitality players and offer convenient services like ticket booking and payment while guaranteeing numerous travellers a good line of communication. Sadly, traditional centralised solutions cannot meet the above demanding requirements. Therefore, a decentralised method is required, which expands the potential in the service-based travel and hospitality industries. Blockchain satisfies tourism's needs by incorporating transactions into an unchangeable distributed ledger [12], which fosters trust [13], transparency [14,15], security [16], and creditability [17]. The uses of blockchain technology are in healthcare [18–20], banking [21,22], education [23,24], IoT [25,26], and governance [27,28]. By enabling direct communication between clients and stakeholders, blockchain technology can replace third-party booking agencies in the tourism industry [9].

The tourism and hospitality industries also carry out transactions using one smart contract connected to the visitor's financial information. Users may now purchase airline tickets [29], hotel rooms [30], and restaurant reservations [31] using a single cryptocurrency

via the same blockchain-based application, ensuring a common view of a unique transactional wallet. Given the importance of this application for facilitating services and creating more efficient and transparent relationships between B to B and B to C in the hospitality business, this study aims to determine the effect of blockchain payment methods on hotel customers' intentions to stay loyal.

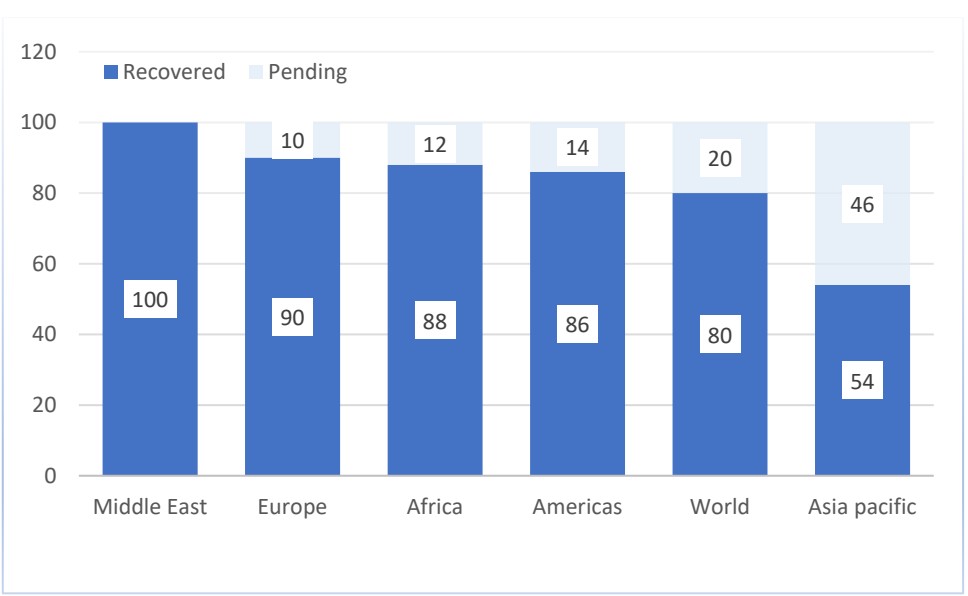

**Figure 1.** Arrivals of foreign tourists: Recovered percentage of 2019 levels in Q1 2023 (%) *. (Source: adapted from UNWTO, * Percentage of Q1 2019 arrivals recovered in Q1 2023 (provisional data)) [11].

## 2. Review of the Literature and Formulation of Hypotheses

### 2.1. Blockchain Payment Services

Blockchain payment services are a brand-new payment processing system that uses blockchain technology to make payments easier [32]. Blockchain is a distributed ledger technology that makes transactions safe, transparent, and unalterable [33]. Due to its ability to lower costs, increase efficiency, and decrease fraud, it is the perfect choice for payment processing. Blockchain is a hard database with a [34] recurrent chain of blocks holding single data transactions transmitted among that specific network's users using a decentralised mechanism [35]. Various sectors such as retail, financial services, supply chain, government and other sectors such as healthcare [36], education, and real estate are using blockchain to facilitate their payment services. Digital payments are transactions that take place using digital technology, such as near field communication (N.F.C.) interactions between an electronic wallet and a cash register or digital currency [37]. Digital platforms are, therefore, "a proprietary or open modular layered technological architecture that supports the efficient development of innovative derivatives, which are embedded in a business or social context" [38]. Blockchain is one example of this kind of platform (Figure 2). Therefore, given the above literature, we hypothesise (H1) that blockchain payment services positively influence the data security perception of hospitality customers.

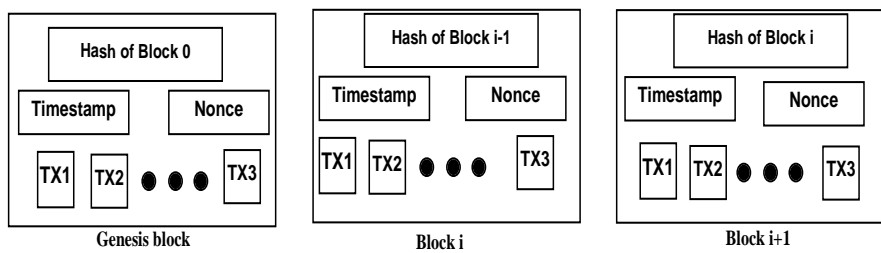

**Figure 2.** Example of a blockchain. Source: Authors' Compilation.

## 2.2. Utilisation Efficiency

Blockchain brings about cost reserve funds with new efficiencies, further developing trust, security, straightforwardness, and the recognizability of information shared through an organisation network [15]. With their consent, members hold a common, changeless record utilised by blockchain for business [39]. Blockchain software platforms utilised in India for paying for transactions and purchasing hospitality services can be integrated into bigger systems, such as high-street banks' foreign exchange payment systems that operate quickly using cryptocurrency. They can also be used to purchase your morning coffee. They are also utilised for direct payment of products and services.

To facilitate payments through the blockchain, many well-known financial services providers, like Visa and Mastercard, are updating their offerings [40]. They also work with various digital asset managers to advance the global payments network. For instance, Mastercard recently introduced its Start Path programme, while Visa recently collaborated with Zipmex to launch products in Southeast Asia.

Many Indian companies launched their blockchain payment services like The "Vajra Platform", a payment system based on blockchain technology, which has been introduced by the National Payments Corporation of India (NPCI) [41], The Rug Republic, HighKart, Purse, Sapna.

It is vital to remember that user adoption, technological literacy, usability, and legal environment may also impact how well hospitality clients use blockchain payment services. These elements influence customers' readiness to embrace and use blockchain payment systems. More studies in hospitality and blockchain payment services can give a deeper understanding of the unique effects on client utilisation efficiency. Therefore, we hypothesise that (H2) data security positively influences the utilisation efficiency of hospitality customers. Furthermore, we hypothesise (H3) that blockchain payment services positively influence the utilisation efficiency of the hospitality customer.

## 2.3. Data Security: Mediating Role

The Internet of Things (IoT) is a network of interconnected mechanical and digital machines [42], computing devices, humans or animals, and objects that can send data across a network without requiring any human or computer contact. Data leakage over the network is a risk throughout the data transmission process [43]. Hence data transfer needs to be safe. Blockchain is a growing collection of records [44,45] or "blocks" that are connected via encryption. Blockchain security is a comprehensive risk management solution for a blockchain network that uses assurance services, best practices, and cybersecurity frameworks to reduce risks against fraud and assaults [46]. This payment method provides a higher level of encryption security, intervention-free functioning, and unchangeable data handling. Blockchain transactions are encrypted, which makes it very difficult for unauthorised parties to read or modify them [47]. Once a transaction is added to the blockchain, it cannot be changed or deleted [48]. This helps to ensure that the data is always accurate and reliable. It is a decentralised system, meaning that there is no single point of failure. This makes it much more difficult for hackers to attack the system [49], as they would need to compromise multiple nodes to succeed. The system's ability to execute transactions without the use of intermediary agents [50] significantly reduces transaction costs. Large service intermediaries like Airbnb, Booking.com, Agora, etc., are predicted to lose some market share by the time blockchain payment solutions reach "maturity" since customers and service providers would likely handle their transactions directly. Consequently, we hypothesise that (H4) data security mediates the effect of blockchain payment services on utilisation efficiency.

## 3. Materials and Methods

### 3.1. Conceptual Model

Most studies on blockchain payment systems use secondary data [51–55]. We conducted this research on Indian territories in South Asia. India's diverse cultures and rich

traditions help to grow the hotel sector. As shown in Figure 3, our study model tends to examine and analyse how tourists would use blockchain payment services.

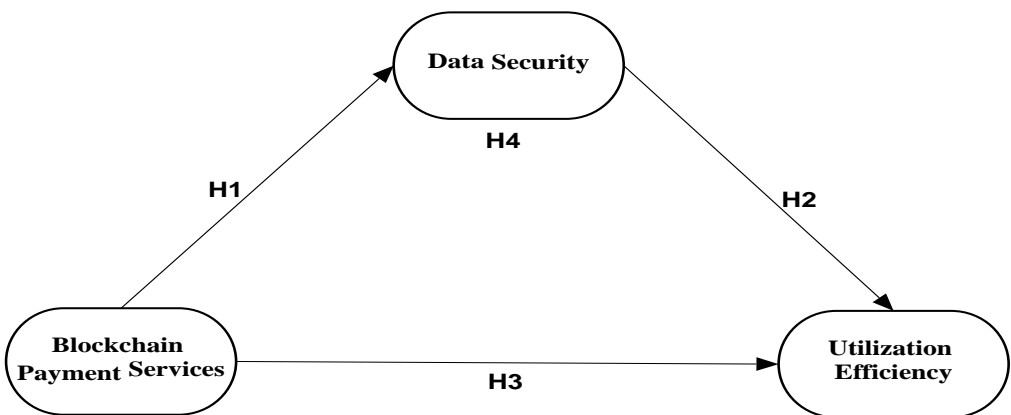

**Figure 3.** Research framework. Source: Authors' Compilation.

### 3.2. Research Instrument

This study investigates the utilisation efficiency of hospitality customer use of blockchain payment services. To accomplish this, we created a questionnaire to gather the required information. In the first section of the questionnaire, demographic information about the respondents was asked for, and in the second section, items for each of the three constructs in our study model were questioned. These constructs are blockchain payment services, data security, and utilisation efficiency. The items used to measure our constructs differed slightly from those used in previous studies since the tests carried out in our hypotheses were slightly different. The blockchain payment services construct was measured with ten items, which were adapted from [9,56,57], data security was measured through five items, which were adapted from [34,58,59], and four items were used to determine utilisation efficiency. Then, each item was rated using a five-point Likert scale ranging from 1 (strongly disagree) to 5 (strongly agree). A specialist group evaluated the first draft of the questionnaire to validate it for clarity and substance legitimacy [60]. The specialists included 5 professionals from the hospitality industry and 10 academics from the faculty of tourism and hotels. They proposed making a couple of humble changes to the questionnaire text.

### 3.3. Data Collection

Data were gathered for this investigation using a random sampling technique. The total number of five- and four-star hotels in Punjab and Haryana is 35 hotels [58]. The study sample data was collected from 25 five-star hotels, where we were permitted to collect the data. The study focused on five and four-star hotels where high-class (national/foreign) customers stay. A structured questionnaire was used to gather data from the respondents.

The general guideline for the S.E.M. technique sample size is that there should be 5–10 times as many instances as there are observable variables [61]. The study hypotheses are connected to a total of 24 variables. Consequently, at least 120 respondents were needed for this study. We, therefore, handed out 20 questionnaires to each hotel manager to distribute to their customers. At the front desk, the managers randomly distributed the 20 questionnaires to hotel customers. This allowed us to retrieve about 60% (301) of the questionnaires distributed for further analysis. We used the two-stage structural equation modelling (S.E.M.) approach [62] in our investigation to determine the suitability of our presented hypothesised (blockchain payment services) model. Confirmatory factor analysis (C.F.A.) was the first step in assessing our suggested model's validity and reliability. The hypotheses we established in Section 2 were then tested by estimating the complete structural model.

## 4. Data Analysis

The descriptive characteristics of the respondents were analysed using SPSS version 22. Cronbach's alpha values were used to test the reliability of the study dimensions. The structural aspects of the conceptualised model were explored using confirmatory factor analysis (C.F.A.) and structural equation modelling (S.E.M.) utilising AMOS version 24 because of the intricacy of the suggested model. All prerequisites for running C.F.A. and S.E.M. were examined and found to be valid. To test the study hypothesis, we then used multiple regression.

### 4.1. Respondents' Demographic

Table 2 shows the total population of respondents by age, gender, marital status, work position, and response rate (a total of 301 valid questionnaires were collected). The fact that 75.4% (227) of the people in our sample were men and 24.6% (74) were female infers that our sample was male-dominated. The demographic analysis showed that 78.7% (237) were married, and 21.3% (64) were single. In addition, in terms of employment status, it was shown that 79.2% (239) were businessmen, 17.3% (52%) were servicemen, and 3.3% (10) were unemployed. Most of the respondents to stay in 4- and 5-star hotels were youths (18–30 years) 49.5% (149), 36.9% (111) were adults (30–60 years), and 13.6% (41) were aged 60 years and above, as presented in Table 2.

**Table 2.** Demographics details.

| Variable | | Total No. | Per (%) |
| --- | --- | --- | --- |
| Gender | Male | 227 | 75.4 |
| | Female | 74 | 26.6 |
| Marital Status | Married | 237 | 78.7 |
| | Single | 64 | 21.3 |
| Age | Youth (18–35) | 149 | 49.5 |
| | Adult (35–65) | 111 | 36.9 |
| | Aged (65–85) | 41 | 13.6 |
| Employment Status | Business | 239 | 79.4 |
| | Service | 52 | 17.3 |
| | Unemployed | 10 | 3.3 |

### 4.2. Confirmatory Factor Analysis

A measurement model's evaluation includes examining the link between latent variables and the variables that represent them [60]. To evaluate the investigation's measuring methodology, C.F.A. was used (Figure 4).

### 4.3. Validity and Reliability

The reliability is measured using Cronbach's method. Table 3 provides the constructs' Cronbach alpha values. Generally, a Cronbach alpha coefficient of 0.7 is required [63–65]. It can be observed that the Cronbach $\alpha$ coefficient for "blockchain payment services" is 0.908, that of "data security" is 0.835, and that of "utilisation efficiency" is 0.880. The measurement of this study is satisfactory in terms of reliability because Cronbach coefficients of all three constructs are greater than 0.7.

Additionally, Table 4 displays the model discriminant validity measurement, which can be used to assess the validity of the convergent and reliable discriminant models. If the AVE of a construct is greater than 0.5 [66] and AVE is greater than MSV, and the square root of AVE is greater than inter-construct correlations, then there is convergent validity for the construct. As shown in Table 4, the AVEs of the three constructs are 0.828, 0.828, and 0.679, respectively, which are all greater than 0.5. It indicates that there is convergent

validity in this study. In addition, we tested the discriminant validity of each concept using the cross-loading measurement criteria [67]. Additional findings, as shown in Table 4, demonstrated that the discriminant validity value was met for each concept.

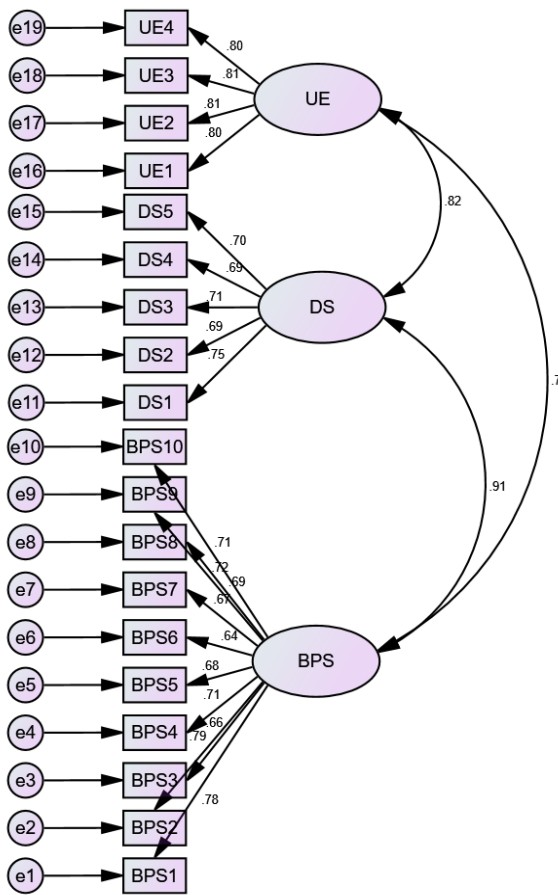

**Figure 4.** Measurement model of the study.

**Table 3.** Reliability of variables.

| Factor | Alpha | Consistency | No. of Items |
|---|---|---|---|
| Blockchain Payment Services | 0.908 | Excellent | 10 |
| Data security | 0.835 | Good | 05 |
| Utilisation Efficiency | 0.880 | Good | 05 |

**Table 4.** Convergent and discriminant validity.

| Variable | CR | AVE | MSV | MaxR(H) |
|---|---|---|---|---|
| | (>0.7) | (>0.5) | (MSW < AVE) | |
| BPS | 0.909 | 0.828 | 0.501 | 0.913 |
| DS | 0.835 | 0.828 | 0.504 | 0.837 |
| UE | 0.880 | 0.679 | 0.647 | 0.88 |

According to Anderson et al. [68], the measurement model, the structural model, the fit-incremental fit index (I.F.I.), the comparative fit index (CFI), the Tucker–Levis Index (TLI), and root mean square error of approximation (RMSEA) estimates must be well above the recommended values by Henseler et al. (2016) [69] and in this study, the C.F.A. presented an acceptable model fit (CMIN = 359.877, $X^2$/df = 2.415; Tucker-Lewis index (TLI) = 0.925; comparative fit index (CFI) = 0.935; incremental fit index (I.F.I.) = 0.935; standardised

root mean square residual (SRMR) = 0.034; root mean square error of approximation (RMSEA) = 0.069, *p* = 0.000; that is, RMSEA was less than 0.08, as shown in Table 5 and Figure 5.

**Table 5.** Model fit indices.

| Model Fit Indices | Criterion | Result |
|---|---|---|
| CMIN | the higher, the better | 359.877 |
| CMIN/DF | ≤3 = accepted fit<br>≤5 = reasonable fit | 2.415 |
| TLI | 1 = perfect fit<br>≥0.95 = excellent fit<br>≥0.90 = acceptable fit | 0.925 |
| IFI | ≥0.90 = acceptable fit | 0.935 |
| CFI | 1 = perfect fit<br>≥0.95 = excellent fit<br>≥0.90 = acceptable fit | 0.935 |
| RMSEA | ≤0.05 = acceptable fit<br>≤0.08 = fit | 0.069 |

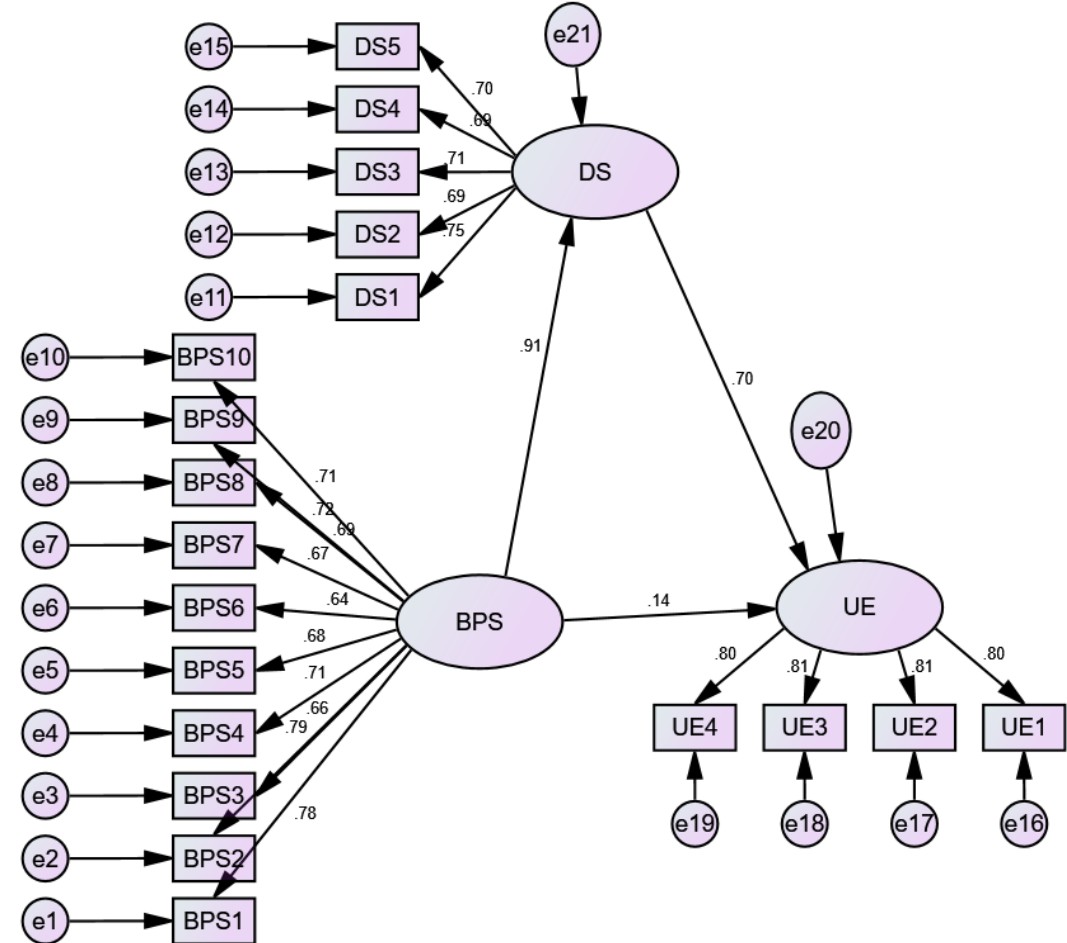

**Figure 5.** Structural model of the study.

*4.4. Hypothesis Testing*

S.E.M. was carried out utilising the information gathered on-site to test the hypotheses. The statistical significance of endogenous components in the study's model structure was revealed by path coefficients of various dimensions [70]. As listed in Table 5, the main factors included the relationship between blockchain payment services and data security

($\beta$ = 0.91, t = 12.714, *p* < 0.001); data security and utilisation efficiency ($\beta$ = 0.70, t = 3.878, *p* < 0.01) were positive and significant paths, blockchain payment services, and utilisation efficiency were insignificant ($\beta$ = 0.14, t = 0.794, *p* < 0.001), as shown in Table 6.

**Table 6.** Hypothesis test.

| Hypothesis Relationship | | | Beta | S.E. | C.R. (t) | *p* | Results |
|---|---|---|---|---|---|---|---|
| H1: BPS | $\longrightarrow$ | D.S. | 0.91 | 0.067 | 12.714 | *** | Supported |
| H2: D.S. | $\longrightarrow$ | UE | 0.70 | 0.172 | 3.878 | *** | Supported |
| H3: B.P.S. | $\longrightarrow$ | UE | 0.14 | 0.153 | 0.794 | 0.427 | Not Supported |

Note: *** *p* < 0.001.

*4.5. Mediation Effect of Variable-Data Security*

We employed the bootstrap approach to examine the mediating impact [71]. Through blockchain payment services, the data security had a full indirect impact on the utilisation efficiency of customers ($\beta$ = 0.638, BC0.95 L.L. = 0.269, and BC0.95 U.L. = 1.253). Moreover, the indirect effects did not cross zero, as proposed by Preacher and Hayes [72], indicating the presence of a mediating influence. The direct impact of blockchain payment services and utilisation efficiency is insignificant (*p* = 0.427). As a result, we may conclude that there was a statistically significant mediating effect, supporting H4 in this research.

**5. Discussion and Conclusions**

Concerning acknowledgement in the movement area, blockchain innovation is still in its earliest stages. Scientists from one side of the planet to the other are endeavouring to foster systems and applications coordinating the utilisation of blockchain in the movement and the travel industry. Quick examination over the most recent few years has expanded blockchain agreeableness, which has led to new and helpful cases from the hospitality industry. Blockchain's combination ensures further developed client information assurance and makes paperless travel encounters conceivable. The article gives perusers significant data about the worth of blockchain innovation in the travel industry and cordiality areas.

The study objective was to research blockchain innovation from the perspective of the lodging area. The review was exploratory and discussed notable papers tending to utilise blockchain in the hospitality industry. By examining the attitudes of hospitality clients towards these applications, this study analysed the role of the blockchain, and applications related to it, in the tourism and hospitality sector. Data security had a significant favourable relationship with blockchain payment services (H1). This supported the previous study by Treiblmaier et al. [54].

Blockchain technology may expedite the identity verification process, customer monitoring, hotel room booking, transportation booking, flight booking, and transaction process in addition to assuring more affordable and quicker payment options, improving the utilisation efficiency of hospitality clients. A consensus that is predetermined by the blockchain's participating members is used to verify transactions on the platform [73,74]. Data security (H2) had a significant positive relationship with customers' utilisation efficiency, showing that the role of data security was impacted by a full mediation between blockchain payment services and the utilisation efficiency of hospitality customers (H4). This corroborates the results of previous research by Kvakarić [75]. Blockchain payment services were insignificant with the utilisation efficiency of hospitality customers (H3). As highlighted by Dam et al. [34] and Salim et al. [76], this showed hospitality customers would not use blockchain payment services without data security. In India's hospitality sector, blockchain payment systems can provide a high level of revenue by establishing high-quality services [77].

*5.1. Implication of the Study*

This study has various theoretical implications. According to the researchers' understanding and the material that is currently accessible, this study is one of the few to look into how effectively customers use blockchain payment services in the Indian hospitality sector. Even though earlier research was conducted in other businesses, like banking, this study is unique and differs from other studies. There are not many research studies that have looked at how data security influences how effectively clients use blockchain payment services in the hospitality sector. Investigating hospitality clients from nations relying on tourism offers significant input to the literature on hospitality. Thus, the findings add to the body of research on client utilisation efficiency and technology-based blockchain payment systems. Finally, by integrating blockchain payment systems into operations and management in the Indian hospitality sector, this study contributes to our understanding of this little-studied phenomenon. The implications of this study support previous studies conducted in various countries, Karim et al. [56], Dogru et al. [78], Khanna et al. [79], and Flecha-Barrio et al. [80]. Hence some of the key implications to consider are listed below.

5.1.1. Enhanced Security

Blockchain technology offers increased security by using cryptographic algorithms to secure transactions [75]. This can reduce the risk of fraud and unauthorised access to sensitive payment information. Managers can focus on improving other aspects of the business without constantly worrying about security breaches.

5.1.2. Reduced Transaction Costs

Blockchain-based payment services can eliminate intermediaries and streamline the payment process. This reduces transaction costs [81] associated with traditional payment methods such as credit cards or bank transfers. Managers can allocate these cost savings to other business areas or offer competitive pricing to attract more customers.

5.1.3. Improved Efficiency and Speed

Blockchain payments are typically faster and more efficient compared to traditional methods that involve multiple parties and manual processing [82,83]. Managers can benefit from faster settlement times, quicker reconciliation, and improved cash flow. This efficiency can also enhance customer satisfaction by reducing waiting times and providing a seamless payment experience.

5.1.4. Increased Transparency

Blockchain technology enables transparent and immutable record-keeping of transactions. Every transaction is recorded on the blockchain, creating a decentralised ledger that can be accessed by authorised parties [56,83]. Managers can leverage this transparency to enhance accountability, improve auditing processes, and build trust with customers and partners.

5.1.5. Integration with Smart Contracts

Blockchain technology can facilitate the implementation of smart contracts, which are self-executing contracts with predefined terms and conditions. This automation can streamline various processes in the hospitality industry, such as room bookings, loyalty programs, and supply chain management. Managers can leverage smart contracts to reduce administrative tasks and improve overall operational efficiency [84].

5.1.6. Data Analytics and Personalization

Insights on consumer behaviour, tastes, and spending patterns can be gained by analysing the massive volumes of transaction data that blockchain payment services create [30]. Managers can utilise these data to personalise marketing efforts, improve customer service, and make data-driven decisions to enhance the overall customer experience.

*5.2. Limitation and Future Study*

The following are the study's shortcomings, which could serve as a foundation for additional investigation. The statistics were mostly collected from patrons of the hospitality sector in the provinces of Punjab and Haryana. The sample size was 301, which did not represent the entire targeted population or viewpoint but was deemed sufficiently representative. To further understand the effects of blockchain payment services in the hospitality industry, future research studies can be undertaken by obtaining more information and expanding the sample size, which calls for visiting more sites. The current study limited the applicability of the findings to the hotel industries by concentrating only on how blockchain payment systems were seen in the efficiency of their utilisation in the hotel sector. Even though blockchain technology has numerous additional uses outside of payments, this study mainly examined general elements of blockchain payments. Future studies might examine brand-new blockchain offerings as monetary and technological remedies for clients in the hospitality sector.

**Author Contributions:** Conceptualization, A.D., S.K., D.R. and K.S.; Methodology, A.D., S.K. and S.G.; Software, A.D.; Validation, S.G.; Formal analysis, S.K., D.R., S.G. and K.S.; Investigation, A.D., D.R. and K.S.; Data curation, S.K., S.G. and K.S.; Writing—original draft, A.D., S.K. and D.R.; Writing—review & editing, S.G. and K.S.; Visualization, S.G.; Supervision, S.G. All authors have read and agreed to the published version of the manuscript.

**Funding:** This research received no external funding.

**Institutional Review Board Statement:** Not applicable in India.

**Informed Consent Statement:** Informed consent was obtained from all subjects involved in the study.

**Data Availability Statement:** Data is available from the 1st Author by request.

**Conflicts of Interest:** The authors declare no conflict of interest.

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
