# Peer review of "Blockchain Payment Services in the Hospitality Sector: The Mediating Role of Data Security on Utilisation Efficiency of the Customer"

_data, 2023_

Round 1

Reviewer 1 Report

Research Design: The research design of the paper appears to be well-structured and appropriate for the study's objectives. The authors aim to investigate the utilization efficiency of blockchain payment services in the hospitality sector and the mediating role of data security. The study utilizes a questionnaire survey to collect data from customers in the hospitality industry. However, it would be beneficial to provide more details on the research methodology, such as the sampling technique and data collection process.

Objectives: The objectives of the study are clearly stated and aligned with the research questions. The authors aim to examine the relationships between blockchain payment services, data security, and utilization efficiency in the hospitality sector. The objectives are relevant and contribute to the understanding of the potential benefits and challenges of blockchain technology in the industry.

Literature: The literature review provides a good overview of the current state of blockchain technology and its applications in various industries. However, it would be helpful to include more recent and diverse references to ensure the coverage of the latest developments in blockchain technology and its specific applications in the hospitality sector. The suggested additional papers could enhance the depth and breadth of the literature review.

Execution: The execution of the study could be further improved. While the demographic information of the respondents is presented, more details about the sample selection process, representativeness, and response rate would strengthen the study's validity. Additionally, it would be valuable to provide a clearer description of the questionnaire items and their development process. Further elaboration on the data analysis techniques employed, such as the specific statistical tests used, would enhance the transparency of the study.

Statistics: The statistical analysis section lacks sufficient information. It is important to provide details on the statistical tests used to test the hypotheses and establish the significance of the relationships between variables. Additionally, the interpretation of the statistical results should be more comprehensive and connected to the research objectives. Providing effect sizes or practical implications of the findings would strengthen the statistical analysis.

Results: The results section presents the main findings of the study, indicating the relationships between blockchain payment services, data security, and utilization efficiency. However, the presentation of the results could be improved by including effect sizes or statistical significance levels. Furthermore, the discussion of the results should be more substantial, linking back to the literature and research objectives, and providing meaningful insights for the hospitality industry.

Suggestions for Enriching the Paper:

  1. Include more recent and diverse references to reflect the latest developments in blockchain technology and its applications in the hospitality sector.
  2. Provide more details on the research methodology, including the sampling technique, sample size, and data collection process.
  3. Elaborate on the questionnaire items and their development process.
  4. Clearly specify the statistical tests used in the analysis and provide effect sizes or practical implications of the findings.
  5. Enhance the discussion section by relating the results to the literature and research objectives and providing meaningful insights for the hospitality industry.

Additional Papers to Consider:

  1. "The Impact of Blockchain Technique on Trustworthy Healthcare Sector" - This paper explores the applications and impact of blockchain technology in the healthcare sector, which can provide insights into the potential benefits and challenges of blockchain adoption in other industries. Sadeq, N., Hamzeh, Z., Nassreddine, G., & ElHassan, T. (2023). The impact of Blockchain technique on trustworthy healthcare sector . Mesopotamian Journal of CyberSecurity2023, 105–115. https://doi.org/10.58496/MJCS/2023/015

  1. "Review on Blockchain Technology: Architecture, Characteristics, Benefits, Algorithms, Challenges, and Applications" - This paper provides a comprehensive review of blockchain technology, covering its architecture, characteristics, benefits, algorithms, challenges, and applications, which can provide a broader understanding of blockchain technology. Vaigandla, K. K., Karne, R., Siluveru, M., & Kesoju, M. (2023). Review on Blockchain Technology : Architecture, Characteristics, Benefits, Algorithms, Challenges and Applications. Mesopotamian Journal of CyberSecurity2023, 73–85. https://doi.org/10.58496/MJCS/2023/012

  1. "Blockchain Technology Applications, Concerns, and Recommendations for the Public Sector" - This paper focuses on the applications and challenges of blockchain technology in the public sector, which can offer insights into the specific considerations and recommendations for implementing blockchain in the hospitality industry. Yaseen, M., Mahadi Bahari, & Omar A. Hammood. (2021). Blockchain technology applications, concerns and recommendations for public sector. Mesopotamian Journal of Computer Science2021, 1–6. https://doi.org/10.58496/MJCSC/2021/001

  1. "Blockchain Technology, Methodology Behind It, and Its Most Extensively Used Encryption Techniques" - This paper discusses the methodology and encryption techniques used in blockchain technology, which can provide a deeper understanding of the technical aspects of blockchain implementation. Mohammed , marwa S., & Hashim, A. N. (2023). Blockchain technology, methodology behind it, and its most extensively used encryption techniques. Al-Salam Journal for Engineering and Technology2(2), 140–151. https://doi.org/10.55145/ajest.2023.02.02.017

  1. "The Genetic Algorithm Implementation in Smart Contract for the Blockchain Technology" - This paper explores the implementation of genetic algorithms in smart contracts for blockchain technology, which can provide insights into innovative approaches for utilizing blockchain in the hospitality industry. Abdul-Sada, H. H., & Furkan Rabee. (2023). The Genetic Algorithm Implementation in Smart Contract for the Blockchain Technology . Al-Salam Journal for Engineering and Technology2(2), 37–47. https://doi.org/10.55145/ajest.2023.02.02.005

None.

Author Response

Dear Esteemed Reviewer

We would like to thank you for the opportunity to resubmit a revised copy of this manuscript.  We would also like to take this opportunity to express our thanks to the reviewers for the positive feedback and helpful comments for correction or modification. We believe this has helped us improve our manuscript. Our responses are noted below. The manuscript has been revised to address your comments.  We hope that these revision have satisfactorily answered your queries and is accepted for publication in Journal.

Sincerely yours

Simon Grima

Submission Id: 2512160

  • Research Design: The research design of the paper appears to be well-structured and appropriate for the study's objectives. The authors aim to investigate the utilization efficiency of blockchain payment services in the hospitality sector and the mediating role of data security. The study utilizes a questionnaire survey to collect data from customers in the hospitality industry. However, it would be beneficial to provide more details on the research methodology, such as the sampling technique and data collection process.

Response: We have taken this suggestion on board and included more information in the methodology to ensure replication. Vide Highlights in section 3.

  • Objectives: The objectives of the study are clearly stated and aligned with the research questions. The authors aim to examine the relationships between blockchain payment services, data security, and utilization efficiency in the hospitality sector. The objectives are relevant and contribute to the understanding of the potential benefits and challenges of blockchain technology in the industry.

Response: Thank you

  • Literature: The literature review provides a good overview of the current state of blockchain technology and its applications in various industries. However, it would be helpful to include more recent and diverse references to ensure the coverage of the latest developments in blockchain technology and its specific applications in the hospitality sector. The suggested additional papers could enhance the depth and breadth of the literature review.

Response: We have taken your suggestions on board and added some current literature. Vide the highlighted references

  • Execution: The execution of the study could be further improved. While the demographic information of the respondents is presented, more details about the sample selection process, representativeness, and response rate would strengthen the study's validity. Additionally, it would be valuable to provide a clearer description of the questionnaire items and their development process. Further elaboration on the data analysis techniques employed, such as the specific statistical tests used, would enhance the transparency of the study.

Response: We have taken your suggestions on board and added more details about the sample selection process, representativeness, and response rate and provided a clearer description of the questionnaire items and their development process. Further elaboration on the data analysis techniques employed. Vide the highlighted sections 3.3 and 4.

  • Statistics: The statistical analysis section lacks sufficient information. It is important to provide details on the statistical tests used to test the hypotheses and establish the significance of the relationships between variables. Additionally, the interpretation of the statistical results should be more comprehensive and connected to the research objectives. Providing effect sizes or practical implications of the findings would strengthen the statistical analysis.

Response: We have taken your suggestions on board and provided a clearer description of the data analysis techniques employed to test the hypotheses and establish the significance of the relationships between variables and added some information to make the statistical results more comprehensive and connected to the research objectives. Vide the highlighted sections 4 and 5.

  • Results: The results section presents the main findings of the study, indicating the relationships between blockchain payment services, data security, and utilization efficiency. However, the presentation of the results could be improved by including effect sizes or statistical significance levels. Furthermore, the discussion of the results should be more substantial, linking back to the literature and research objectives, and providing meaningful insights for the hospitality industry.

Response: We have taken your suggestions on board and related the results to the literature and research objectives and providing meaningful insights for the hospitality industry . Vide the highlighted sections 4 and 5.

  • Suggestions for Enriching the Paper:
  1. Include more recent and diverse references to reflect the latest developments in blockchain technology and its applications in the hospitality sector.

Response: We have taken your suggestions on board and added some current literature. Vide the highlighted references.

  1. Provide more details on the research methodology, including the sampling technique, sample size, and data collection process.

Response: We have taken this suggestion on board and included more information in the methodology to ensure replication. Vide Highlights in section 3.

  1. Elaborate on the questionnaire items and their development process.

Response: We have taken your suggestions on board and provided a clearer description of the questionnaire items and their development process.

  1. Clearly specify the statistical tests used in the analysis and provide effect sizes or practical implications of the findings.

Response: We have taken your suggestions on board and provided further elaboration on the data analysis techniques employed. Vide the highlighted sections 3.3 and 4.

  1. Enhance the discussion section by relating the results to the literature and research objectives and providing meaningful insights for the hospitality industry.

Response: We have taken your suggestions on board and related the results to the literature and research objectives and providing meaningful insights for the hospitality industry . Vide the highlighted sections 4 and 5.

  • Additional Papers to Consider:
  1. "The Impact of Blockchain Technique on Trustworthy Healthcare Sector" - This paper explores the applications and impact of blockchain technology in the healthcare sector, which can provide insights into the potential benefits and challenges of blockchain adoption in other industries. Sadeq, N., Hamzeh, Z., Nassreddine, G., & ElHassan, T. (2023). The impact of Blockchain technique on trustworthy healthcare sector . Mesopotamian Journal of CyberSecurity, 2023, 105–115. https://doi.org/10.58496/MJCS/2023/015

 Response: Added

  1. "Review on Blockchain Technology: Architecture, Characteristics, Benefits, Algorithms, Challenges, and Applications" - This paper provides a comprehensive review of blockchain technology, covering its architecture, characteristics, benefits, algorithms, challenges, and applications, which can provide a broader understanding of blockchain technology. Vaigandla, K. K., Karne, R., Siluveru, M., & Kesoju, M. (2023). Review on Blockchain Technology : Architecture, Characteristics, Benefits, Algorithms, Challenges and Applications. Mesopotamian Journal of CyberSecurity, 2023, 73–85. https://doi.org/10.58496/MJCS/2023/012

 Response: Added

  1. "Blockchain Technology Applications, Concerns, and Recommendations for the Public Sector" - This paper focuses on the applications and challenges of blockchain technology in the public sector, which can offer insights into the specific considerations and recommendations for implementing blockchain in the hospitality industry. Yaseen, M., Mahadi Bahari, & Omar A. Hammood. (2021). Blockchain technology applications, concerns and recommendations for public sector. Mesopotamian Journal of Computer Science, 2021, 1–6. https://doi.org/10.58496/MJCSC/2021/001

 Response: Added

  1. "Blockchain Technology, Methodology Behind It, and Its Most Extensively Used Encryption Techniques" - This paper discusses the methodology and encryption techniques used in blockchain technology, which can provide a deeper understanding of the technical aspects of blockchain implementation. Mohammed , marwa S., & Hashim, A. N. (2023). Blockchain technology, methodology behind it, and its most extensively used encryption techniques. Al-Salam Journal for Engineering and Technology, 2(2), 140–151. https://doi.org/10.55145/ajest.2023.02.02.017

 Response: Added

  1. "The Genetic Algorithm Implementation in Smart Contract for the Blockchain Technology" - This paper explores the implementation of genetic algorithms in smart contracts for blockchain technology, which can provide insights into innovative approaches for utilizing blockchain in the hospitality industry. Abdul-Sada, H. H., & Furkan Rabee. (2023). The Genetic Algorithm Implementation in Smart Contract for the Blockchain Technology . Al-Salam Journal for Engineering and Technology, 2(2), 37–47. https://doi.org/10.55145/ajest.2023.02.02.005

 Response: Added

Reviewer 2 Report

Title: “Blockchain payment services in the hospitality sector: the mediating role of data security on utilization efficiency of the Customer”

 The article is prepared on a current topic; the article is of interest to scientists; and the article can be published, but it needs preliminary thorough revision. The article is devoted to the research how blockchain payment methods affect hotel Customers' intentions to stay loyal.

Notes and recommendations for authors:

1) in my opinion, it is worth specifying the research question at the beginning of the article

2) the data presented in the tables and figures should be described and interpreted in more detail in the text of the article

3) I think it is worth making two separate sections: Results and Discussion. In the section Results, the authors’ analysis of the research results should be presented. In the section Discussion, the results obtained by the authors should be compared with previous studies.The section Discussion should be developed by supplementing it with links to specific studies and publications by scientists who have studied similar research problems

Author Response

Dear Esteemed Reviewer

We would like to thank you for the opportunity to resubmit a revised copy of this manuscript.  We would also like to take this opportunity to express our thanks to the reviewers for the positive feedback and helpful comments for correction or modification. We believe this has helped us improve our manuscript. Our responses are noted below. The manuscript has been revised to address your comments.  We hope that these revision have satisfactorily answered your queries and is accepted for publication in Journal.

Sincerely yours

Simon Grima

Submission Id: 2512160

The article is prepared on a current topic; the article is of interest to scientists; and the article can be published, but it needs preliminary thorough revision. The article is devoted to the research how blockchain payment methods affect hotel Customers' intentions to stay loyal. 

Notes and recommendations for authors:

1)in my opinion, it is worth specifying the research question at the beginning of the article

 Response:

The aim of this study has been added to the introduction- vide highlighted. The article makes 4 hypothesis:

Hypothesis.1 (H1): Blockchain payment services positively influence data security perception of hospitality customer.

Hypothesis.2 (H2): Data security positively influences the utilization efficiency of hospitality customer

Hypothesis.3 (H3): Blockchain payment services positively influence the utilisation efficiency of hospitality customer

Hypothesis.4 (H4): Data security mediates the effect of blockchain payment services on utilization efficiency

These follow the literature from which they are derived.

2) the data presented in the tables and figures should be described and interpreted in more detail in the text of the article.

Response: We have taken your suggestions on board. The highlighted section 4 explains the models and tests used for the analysis of the data, while the rest of the section explain the results in the tables and figures.

3) I think it is worth making two separate sections: Results and Discussion. In the section Results, the authors’ analysis of the research results should be presented. In the section Discussion, the results obtained by the authors should be compared with previous studies.The section Discussion should be developed by supplementing it with links to specific studies and publications by scientists who have studied similar research problems

Response: We have taken your suggestions on board. Results are presented in section 4 and Discussion and conclusions and implications are described in section 5